# Advanced Waveguide Based LOC Biosensors: A Minireview

**DOI:** 10.3390/s22145443

**Published:** 2022-07-21

**Authors:** Muzafar A. Kanjwal, Amal Al Ghaferi

**Affiliations:** Mechanical Engineering Department, Khalifa University, Abu Dhabi 127788, United Arab Emirates; muzafar.kanjwal@ku.ac.ae

**Keywords:** mid-infrared, waveguides, LOC devices, graphene incorporation

## Abstract

This mini review features contemporary advances in mid-infrared (MIR) thin-film waveguide technology and on-chip photonics, promoting high-performance biosensing platforms. Supported by recent developments in MIR thin-film waveguides, it is expected that label-free assimilated MIR sensing platforms will soon supplement the current sensing technologies for biomedical diagnostics. The state-of-the-art shows that various types of waveguide material can be utilized for waveguide spectroscopic measurements in MIR. However, there are challenges to integrating these waveguide platforms with microfluidic/Lab-on-a-Chip (LOC) devices, due to poor light–material interactions. Graphene and its analogs have found many applications in microfluidic-based LOC devices, to address to this issue. Graphene-based materials possess a high conductivity, a large surface-to-volume ratio, a smaller and tunable bandgap, and allow easier sample loading; which is essential for acquiring precise electrochemical information. This work discusses advanced waveguide materials, their advantages, and disease diagnostics with MIR thin-film based waveguides. The incorporation of graphene into waveguides improves the light–graphene interaction, and photonic devices greatly benefit from graphene’s strong field-controlled optical response.

## 1. Introduction

The human body contains various fluids, for instance, blood, urine, semen, vaginal secretions, saliva, synovial fluid, tears, and gastric fluids, that carry critical details of health conditions. In addition, these fluids provide an alternative approach for disease identification and disease prognostication [1]. The investigation of bio-fluids to obtain precise data for clinical diagnosis is of great significance. It is a well-known fact that a high percentage of the world population is experiencing various kinds of disorders and disabilities; therefore, general public health is subjected to a substantial burden of disease diagnosis and medical attention. Inexpensive medical tests are essential for the examination of the general public for disease identification and also for drug discovery [1,2]. There are various approaches available to investigate bio-fluids; for instance, spectroscopic methods formulated on Raman and infrared absorption, microscopy, fluorescence techniques, and chromatographic approaches. Each method is unique and has its benefits and limitations, and can work in synergy for better results [2].

The word mid-infrared (MIR) usually signifies the spectral range from around 4000 cm^−1^ to 400 cm^−1^ (2.5–25 µm). This spectral range permits qualitative and quantitative spectroscopic findings that include an elemental, structural, and configurational understanding of molecular components. Mid-infrared spectroscopic technique can be used for solid, aqueous, and gaseous samples. Essentially, important information is acquired based on the excitation of basic vibrational, rotational, and vibro-rotational transitions, yielding a spectrum (a molecular fingerprint) for the molecule under study. Specifically, because of their resonances, molecular species are characterized by a particular MIR spectrum, therefore allowing IR spectroscopy to become a regular investigation tool for recognizing and characterizing these molecules. Normal waveguides established on spectroscopic methods, internal reflection, and attenuated total reflection spectroscopy (ATR) were reported by Harrick and Fahrenfort in the 1960s [3,4]. The development of MIR transparent optical fibers, specifically in late 1980s and early 1990s, involved materials such as chalcogenide, a perspective material for optoelectronics, with a high optical transparency in the infrared (IR) region; silver halides (AgX); sapphire; and hollow waveguide (HWG) structures, which played a critical role in the transformation of conventional IR spectroscopic methods into fiber optics-based MIR sensing applications [5]. The property of molecular selectivity enables MIR spectroscopic techniques to function as feasible methods for prompt label-free investigations, which facilitated the establishment of optical sensing technologies. Effective sensing and surveillance platforms have been introduced in healthcare [6,7], for commercial applications [8,9], and in atmospheric examinations, such as the investigation of greenhouse gases [10,11]. Previously, MIR spectroscopy has been regarded as a laboratory fascination, with restricted usage in real-world events. However, with the rising acceptance of nano/microfabricated optical constituents that include light sources, and waveguides that also serve as sensing materials, normal instrumentation and MIR-based sensing platforms can considerably improve their physical characteristics, without giving up of their sensitiveness, hardness, authenticity, and selectiveness [12,13]. The advancement of waveguide technology has enabled advanced MIR sensing technologies to exploit the established fundamental concepts of waveguides, especially the investigation of actively absorbed materials or selected components in matrices, such as water molecules, that restrict the utilization and application of normal MIR spectroscopy in transmission methods. Infrared attenuated total reflection spectroscopy (IR-ATR) is one of the most appropriate measuring technologies for dealing with this kind of complicated analytical situation, [3] and has gained advantages from the development in MIR-based waveguide technology.

The utilization of optical waveguides has various benefits in relation to conventional liquid absorption spectroscopy, including easy functioning, better sensitivity for minute sample sizes, quick on-chip incorporation with microfluidics, and miniature representation. The human body is composed of around 60% water; therefore, biochemical reactions occur in an aqueous medium. This causes major problems for MIR spectroscopy, as many aqueous solutions strongly interact with light at certain MIR wavelengths. The research on thin-film waveguide platforms for MIR spectroscopy is in its inception and is supported by very little evidence. This paper helps to understand the trailblazing MIR materials and waveguide spectroscopic evaluation of bio-fluids for clinical diagnostics. Biosensors established on thin-film waveguides present transportable, firm, and reliable detection sensing techniques. Utilizing microfabrication mass manufacturing, a number of biosensors can be incorporated on a wafer, therefore permitting the assimilation of electronics and microfluidics on the same bio-chip. Although MIR spectroscopic evaluations of conventional waveguide absorption spectroscopy have mostly investigated liquid samples, waveguide biosensors have a tremendous significance when investigating ultrafine bio-layers, where a powerful relationship between light and bio-layers can be regulated over the entire surface of the waveguide. These waveguides can be modified by inserting designed bends and coils over multi-centimeter stretches [14,15], and their acuteness can be enhanced by introducing dangling structures, where strong evanescent waves reside on the exterior of the system, while the central waveguide can be enclosed by biofluid/analyte.

MIR spectroscopy uses a Fourier transform infrared (FTIR) spectrometer in liquid samples, with a specimen in cuvettes composed of IR active crystalline material, with white light. The MIR stretching is influenced by the dominant water absorption background, making cuvettes hard to operate due to clogging properties. This issue can be addressed by utilizing attenuated total reflection (ATR) crystals incorporated with FTIR; as this technique involves evanescent field, it experiences less intervention of biofluids, for instance, dispersion from living cells [2].

The waveguide is an integral part of contemporary technology. Waveguide evolution is essential, or else the overall advancement of optical technologies would suffer [16]. Substrates such as silicon, germanium etc. can produce a wide bandwidth and are essential for accomplishing better results [17,18,19,20]. However, the utilization of these substances is impeded due to a narrow bandwidth, higher energy utilization, and high cost, etc. [21,22].

Several photonic platforms have been suggested for chemical or biosensing in the midinfrared (MIR) region, owing to their direct label-free investigation potential, emerging from the distinctive characteristic spectra of the active molecular species under investigation (a.k.a. molecular fingerprint) [23,24,25,26,27,28]. Out of these, waveguide-established technology yields an interesting method for miniaturizing sensors and on-chip incorporation of different units, such as microfluidics [29,30], light sources [31,32], light-sensitive detectors [33], and optronic electric circuits [34,35]. In the recent past, numerous MIR material systems have been studied for use in possible chemical or biological sensing operations [36,37,38], which includes silicon (Si) slots or optical waveguide for ethyl alcohol and ethyl nitrile observation [39], chalcogenide waveguide for methyl hydride and nitrous oxide sensing [40], and germanium-on-silicon (GOS) for methylbenzene observation [41]. Silicon-on-insulator (SOI) waveguides allow an additional and interesting method for sensing applications, which can take advantage of complementary metal oxide semiconductor (CMOS) or complementary-symmetry metal oxide semiconductor fabrication process and abundant frameworks [42,43]. Nevertheless, the extreme absorption due to the buried oxide (BOX) inhibits the utilization of SOI waveguides for the long-wave infrared (LWIR) wavelength region over 6 µm, which involves the vibrational impression of abundant covalent bonds, such as C–H, O–H, N–H, etc. [44,45]. Moreover, a subwavelength grating (SWG)-aided suspending Si waveguide has been found to be a feasible option for resolving this issue and can completely utilize the transparency window of Si approximately at 8 µm [46,47]. In the recent past, one research group analytically exhibited the viability of suspended Si waveguides with a transmission window of 7.67 µm and a path loss of 3.1 ± 0.3 dB/cm [48]. Limit of detection (LoD) and response time are critical parameters of gas/liquid sensors for on-spot instant detections. Up to now, the normal limits of detection (LoDs) of waveguide technology have been detected from as low as a few hundred ppm to as high as tens of thousands ppm [49,50,51,52,53]. The enhancement of the sensitiveness, due to the LoD, is extremely beneficial for meeting the needs of real-world applications. To increase the LOD, certain covering layers have been introduced for the sample aggregation and sensitivity improvement of waveguide technology [54,55,56,57]; normally, they lower reaction times by few seconds, or up to few minutes.

Two-dimensional (2D) materials, such as graphene, have been of great interest for operations in electronics and photonics, due to their exceptional characteristics [58,59]. Graphene is generally described as an outstanding plasmonic compound for light–material relationships [60]. Due to the direct distribution of Dirac fermions, graphene can become assimilated with different substrates, to produce waveguides, sensors, detectors, polarizers etc. [61,62,63]. In the optic-region, graphene-based waveguides play a very important part in integrated optical circuits, detectors, and LOC devices.

## 2. Fundamental Principles and Designing of Micro-Optic Components and Production

As mentioned in the literature, calorimetry and scattering assessment are studied longitudinally, along the major portion of the sample. The dispersion of light along the sample results in a weakening of the intensity that corresponds to the optical path extent l (cm^−1^) and sample concentration c (M); and is represented by the Beer–Lambert law [64]:$$A\lambda =-\log_{10}\left\{{I|I}_{o}\right\}=\varepsilon \left(\lambda \right)\mathrm{cl}$$
where I_o_ represents the intensity evaluated along l without any analyte; on the contrary, I designates the intensity evaluated along the similar optical path in the presence of the analyte; and ε is the molar attenuation coefficient, which changes as an operation of the wavelength. It is important to mention that the attenuation of the sample is denoted as absorbance A with absorbance units in (A.U.). The analyte is the main factor that causes the attenuation of incident rays. Appropriate solvent and reflection at interfaces also contribute to the attenuation of light. For precise evaluation, I_o_ is determined with the same optical arrangement and other conditions as utilized in the evaluation of I.

The design of micro-optic components follows the fundamental laws of optics, as the diffusion limit is not attained. Therefore, the following hypothesis should be investigated. As a matter of fact, the easiest way to integrate light sources into LOC is pigtailed fiber optics, in which the various light origins, photosensors, and mass spectrometer are connected [65,66]. Therefore, if micro-optic components are devised in agreement with fiber optics, they can be easily incorporated in LOC as a fundamental component and do not depend on any specific operations. Regarding the optical configuration, it is important to note that light rays are angled at an intersection through two components in accordance with Snell’s law,
(1)$$n_{1}\sin\theta_{1}=n_{2}\sin\left(\theta_{2}\right)$$
where n_1_ and n_2_ are the corresponding indices of refraction of adjoining components and θ_1_ and θ_2_ are the different incident angles, before and after going through the intersection.

Another very important formula is the lens-maker’s equation [66].
(2)1f=(n1−1) [1R1−1R2+(n1−1)dR1R2]
where f represents the focal distance, R_1_ and R_2_ represent the radius of curvature, and d implies the thickness of the optic lens. Therefore, by aligning different parameters such as circular intersection (as shown in Figure 1a), the well-known indices of refraction, dispersion, and light configuration can be adapted to fit the particular conditions of the LOC implementation. The most important factor that is needed for controlling the light movement is an intersection connecting two substances of non-identical indices of refraction. Air is generally used as an important and dynamic substance for distinguishing many optical components. Figure 1b displays one such illustration of how this optical process can be exploited in a LOC; the combination of a pair of collimating lenses, light and sample relationship, and a smooth horizontal air–glass interface increase the optical path twofold across the specimen.

### Infrared Spectroscopy and Clinical Discovery

Infrared spectroscopy is an influential analytical tool for understanding various kinds of clinical conditions. Synchrotron infrared integrated with FT-IR microscopy, known as synchrotron radiation-based Fourier transform infrared (sr-ftir) spectro-microscopy, is a new and evolving bioanalytical and imaging device. SR-FTIR has exhibited great potential for diagnostic studies, where the photon density is 10^2^–10^3^ times more than the conventional Globar photon flux density, permitting better resolution of an optical imaging system and a better signal-to-noise ratio [68]. Therefore, (SR-FTIR) may be used for the inspection of biomaterials [69] and has exhibited remarkable clinical applications in examining various diseases [70,71], e.g., cancer [72] and in cell biology research [73,74]. SR-FTIR may play a critical role in clinical studies and may help to understand forthcoming clinical determinations and investigations with traditional scientific procedures.

In this regard, a research group [75] studied and employed synchrotron FT-IR to study the chemical reactions of astrocytes connected to gene variants of the oxidoreductase enzyme copper, zinc superoxide dismutase, involved in familial amyotrophic lateral sclerosis (ALS), and accounting for 5 to 10% of all ALS cases, and which is a progressive neurodegenerative disease that results in muscle weakness, disability, and, finally, death Figure 2.

These researchers employed (SR-FTIR) to comprehend the biochemical changes in motor neuron disorder and established a confined intensification of ν_s/as_(CH_2_), ν_s/as_(CH_3_), and ν(C=O) bond vibrations and explained the higher accumulation of phospholipids in the middle area of ALS astrocytes and the substantial phospholipid acyl concentration, which were absent in nontransgenic littermate (Ngs) model astrocytes [75,77,78]. The above-mentioned spectroscopic impression permitted distinguishing the immense accumulation of phospholipid vesicles and superoxide dismutase 1 enzyme, oxidative degradation of lipids, and metabolic choline regulations; these allowed subsequent biomarker location and the evolution of unique diagnostic policies. Another research group demonstrated that (SR-FTIR) has the ability to introduce spectroscopic molecular biomarkers for developing stem cells for medical operations [79].

It is fair to say that IR spectroscopy has tremendous importance in the medical field and is suitable for all biological specimens; delivering, high-class information that helps in critical clinical resolutions. The progress mentioned earlier may have a great impact on healthcare and can clarify contemporary outlooks, may help to reduce death rates and morbidity, and may improve the wellbeing of people around the world. The addition of bifunctional quantum cascade laser/detector QCL to IR technology allows the utilization of non-liquid nitrogen-based detectors. IR has scientific significance and is ready for clinical adaptations. The specific evolution discussed above and enterprising attitudes towards spectroscopy are needed, in order to launch products onto the market. This would need the very close association of researchers, physicians, managers, and investors, to ensure clinical care and patient satisfaction [77].

## 3. Discussion

### 3.1. First-Edition MI Waveguides

The dispersion of light radiations within the waveguide structure at a wavelength greater than internal reflection elements (IRE) results in the development of evanescent waves at certain spots, corresponding to every reflection across the IRE surface. The IRE thickness plays an important role, and the total number of internal reflections is indirectly proportional to the waveguide proportions, due to the geometrical properties. These properties result in increasing the sensitivity and reducing the waveguide density. However, decreasing the IRE density improves the sensitiveness significantly, and causes less photon interaction with IRE. The whole process has a negative impact on the signal-to-noise (S/N) ratio and requires additional detectors for proper functioning. The (S/N) ratio also depends on the refractive index (RI) of the waveguide material used, while decreasing the RI of materials enlarges the penetration depth and results in a better relation of sample molecules at the waveguide intersection. In general, regulating the key physical parameters such as waveguide composition, proportions, and geometric design, as well as optical features such as evanescent waves, are crucial factors for obtaining the best results in sensing applications.

### 3.2. Second-Edition MI Waveguides

In recent times optical fibers being utilized in spectroscopy and sensing platforms has emerged as potential replacements for regular IRE in spectroscopic techniques, exploiting the benefits of MIR optical fibers as waveguide platforms [80]. The utilization of optical fibers as sensing components allows the segregation of real sensing intersections and the detector, promoting optimum working conditions of optical fiber sensors for on-spot investigations. The multi-meter length of MIR optical fibers and their mechanical strength are important parameters for these kinds of waveguides. The major benefits of second-generation waveguides over conventional waveguides are their ability for proper light directing and precise sensing in isolated experimental setups, such as for directing photons to the test specimen. In this class of waveguides, the active detection zones of optical fibers (the point of interaction) can be amended and these active zones across fibers may be enhanced in relation to the first-generation class. The orientation of active zones can be regulated for optimizing internal reflections. Moreover, based on the rotational symmetry of optical fibers, evanescent waves are utilized to produce the absorption signal that is estimated, whereas in traditional IRE, one side is exploited in the estimation processes. To enhance the sensitivities of these classes of waveguides, a tapering method has been introduced. This method produces fiber optics of certain micrometers and has mechanical strength issues. Therefore, these kinds of optical fibers are difficult to use in modern sensing applications. To address these issues, the development of third generation waveguides is essential.

### 3.3. Third-Edition MI Waveguides

The dimensions of the waveguide structure are important characteristics. The total number of reflections is indirectly dependent on the thickness of the waveguide material. Decreasing the thickness results in increasing the wavelength, and eventually homogenous evanescent waves are created across the waveguide surface, resulting in a significant increase of sensitivity [81]. Therefore, the optical dispersion and total internal reflections are not significant in these thin-film waveguides, and the produced evanescent waves are expressed by advanced prototypes established on wave investigations. The waveguides accomplished with these proportions are usually known as third generation waveguides or thin-film waveguides. It is important to mention that thin-film waveguides or integrated optical waveguides (IOWs) enhance the evanescent wave strength and allow the precise development of waveguide designs for a particular operation. These types of thin-film waveguides or (IOWs) are normally placed on a substrate of appropriate refractive index, providing better mechanical strength and precise sensing platforms.

With the evolution of thin-film waveguides and MIR spectroscopy, methods primarily used for silicon microfabricating are being embraced in the MIR spectroscopic lab-on-chip applications. Methods such as electron beam patterning and photolithography expedite the smooth functioning of even complex optical devices [82]. It is obvious that contemporary waveguide established MIR techniques exploit waveguides as dynamic transducers, instead of only functioning as a light radiation transmission duct. This is a very important factor according to an spectroscopic outlook. A transducer perform crucial functions as a sensing element to produce the analytic signal, by allowing consistent communication between incident light and specimen units

Contemporary optical molecular sensors or chemosensors utilize the influence of the particular molecular impression produced in the MIR region, where as a small number of optical biosensors function in the MIR region, as reported in the literature. The major reason for this is that the sensitivity of normal IRE and the established optical fiber MIR sensors are inadequate for identifying small amounts of important biological molecules, e.g., nucleic acids or amino acids in small quantities. These issues in first- and second-generation waveguides are being addressed by third generation waveguides, in which the low sensitivity is easily alleviated by incorporation of suitable polymeric membranes, such as sol-gel, precisely at the thin-film waveguide surface, functioning as enrichment membranes, an approach relevant for traditional IRE, optical fibers, and advanced IOWs. Therefore, observation quantities at the parts per billion (i.e., mg/L) concentration level were achieved with this type of sensing platform. Utilizing this sensing regime, IR-ATR-established chemosensors were devolved, performing significantly in environmental pollutant observation [83,84] and biological operations [85].

As is well reported in the literature, there are less materials available for organizing lab-on-chip waveguide systems. The selected semiconductor materials must possess appropriate physicochemical properties, such as optimum temperature, pH, and mechanical properties. In addition, the introduction of modern QCL and ICL light origin, requires compatibility between the waveguide surface and materials used, to take advantage of the promising features of these excellent light sources.

## 4. The Significance of Thin-Film Waveguides

An ATR component that is used in practice is an multi-modal waveguide of a certain millimeter thickness. The incoming rays are captured by combining inner reflections and those deflected off the vertex at certain specific spots, where the interaction between a total evanescent field with aqueous bio-samples comes into play, as shown in Figure 3a. Every reflection results in absorption, due to the interaction of the evanescent field in the bio-samples; therefore, multiple reflections and an exterior range are important requirements to obtain an efficient absorption and significant spectroscopic impressions. Thin-film waveguides are basically composed of very fine ATR components, where distinct reflections result in the propagation of the evanescent field across the waveguide upper layer (Figure 3b). It can be seen from Figure 3b that the waveguides consist of an upper engineered layer with a sharp index of refraction that lies on a less refractive index substance underneath, and which spreads the evanescent field to the top layer. This spreading evanescent field results in a better absorbance in relation to the well-established ATR constituents. Moreover, such types of waveguide are configurated in a way where it is possible to regulate the absorption and, consequently, noise.

The focal point of this section is the contemporary field of thin-film waveguide spectroscopic technology in MIR, which is critical for the lab-on-a-chip incorporation of MIR sensing structures and eventually IR-lab-on-a-chip technologies.

### 4.1. Thin-Film Waveguides

The ATR absorption spectroscopic technique in the MIR spectroscopic range is being regularly utilized for the investigation of materials of interest, due to the exploitation of a well-known phenomenon called total internal reflection (TIR). The TIR phenomenon comes into play when the reflected light at a certain angle of incidence eclipses a certain critical value or angle, e.g., θ_c_ = sin^−1^(n_c_/n_wg_) at the intersection of a waveguide and the adjoining environment. If the waveguide has a higher refractive index with respect to the adjacent medium, then segments of the evanescent field enter into the adjoining surroundings at a diminishing field strength across the waveguide upper layer. If sample particles are available at the waveguide upper layer or inside the penetration depth (dp) (penetration depth is an evaluation of how much light rays or any electromagnetic waves can infiltrate into a material of interest, to approximately 37% of its initial value below the surface), they can react with the evanescent waves produced at an intersection, which results in attenuation of the generated evanescent field, and eventually results in the production of an IR-ATR active electromagnetic spectrum [86]. The growing strength of the evanescent waves influences the attainable signal-to-noise ratio (SNR) during the absorption evaluation process. The strength of evanescent waves is directly proportional to the relative permittivity (dielectric constant) at the intersection, which is also dependent on the participating matter, dimensions, and transaction area of the thin-film waveguide. The increase in noise during the evanescent wave estimation actually regulates certain important factors, such as waveguide design, configuration, and density, to increase the spectroscopic sensitivity.

So far, the propagating evanescent wave has been regarded as one-directional, as described by the dimensions of the planate waveguide system (slab waveguide) (Figure 4a). In addition, this propagating wave can obliquely diffuse inside the waveguide surface. The propagating wave may also become horizontally captured, which results in an enhancement of the optical strength [87]. Figure 4 graphically represents the well-known waveguide systems that permit different modal conducts in waveguide structures, as in common slab waveguides (Figure 4a–d). There are several methods that promote the production of such waveguide systems. Conventionally, a material of higher refractive index is mounted on a lower refractive index material or the previously placed waveguide upper layer is engraved and the strip waveguide form is maintained (Figure 4b). If the engraving is ceased prior to lower layer attainment, this results in rib waveguide formation, which generally active in the visible region. It is important to mention that the modal conduct depends on the rib depth and not on the waveguide dimensions (Figure 4c). If an upper cladding film is available in the middle of the lower substrate layer, as well as the existing one, a ridge waveguide can be formed (Figure 4e). An embedded waveguide or buried waveguide is formed by enhancing the index of refraction of a lower substrate layer (Figure 4f). Fiber optics is frequently utilized in the class of waveguides employed in the telecom industry and other fields where the incorporation of light into waveguides is required [88,89]. 

### 4.2. Advanced Technology of Thin-Film MIR Waveguides

The most recent development in waveguide technology indicates that various materials could be exploited in the waveguide manufacturing process. Some important matters must be addressed when selecting suitable materials for waveguide creation. The materials used must have an appropriate durability, suitable clarity, chemical resistance, a higher index of refraction, fabricability, and higher SNR. MIR spectroscopic technology has determined various waveguide structures, using normal aqueous media to set up basic principles and processes, prior to examination of biofluids for clinical diagnostic operations.

### 4.3. GaAs/AlGaAs Waveguides

The state of the art shows that there are several recent deposition methods available for the production of very fine semiconductor films or layers (e.g., GaAs, AlGaAs). These procedures include molecular beam epitaxy (MBE) and metal-organic vapor phase epitaxy (MOVPE), for fabricating materials of distinct proportions and configurations. The MBE technique was utilized to fabricate the initial GaAs/AlGaAs waveguide systems. These thin-film systems were developed on a Si-doped GaAs underlayer having a refractive index of 2.8, as shown in Figure 4. The 6-μm Al_0.2_Ga_0.8_ optic coating with a refractive index of 3.2 was developed using the epitaxial method, and succeeded by the original waveguide surface with a refractive index 3.3. Figure 4 also shows the information acquired and shows the procurement of a single-mode outline across the *x*-axis. These thin-film slab waveguides encourage the captivity of MIR emissions at 90° to the propagation axis and result in mode fluctuations and lateral captivity.

Similarly, 2-nL drops were placed on a 200-μm broad waveguide surface using microcapillaries having a narrow-pointed top. A mixture of acetic anhydride in a low volatile solvent, diethylene glycol monoethyl ether (DGME), was used for calibration. The evanescent wave’s incorporation and biosensor reactivity to more droplets being accumulated over the ridge waveguide surface is shown in Figure 5.

The research groups of Mizaikoff and Faist jointly reported the first thin-film waveguides developed from GaAs/AlGaAs using two techniques previously mentioned, yielding a spectroscopic transmission factor of 13 μm [12]. These waveguides have tremendous importance for on-chip incorporated spectroscopic sensing techniques.

These semiconductor thin-film waveguides implement the capture of spectroscopic waves at right angles to the main axis, and this alters the modal outline obliquely along the radiation distribution track. To increase the analytic reactivity in evanescent wave assimilation investigations, GaAs ridge waveguides were micro-structured at different stretches [87]. For micro-structuring, a light-sensitive material was deposited on the wafer utilizing spin coating technology and this was followed by exposure to UV rays. This process was followed by transferring these light-sensitive waveguide strips to the wafer using an etching technology. Subsequently, this material was detached and the wafers were split using a surgical blade to obtain waveguide chips.

Recently, a much more advanced actuator device established on GaAs/AlGaAs was introduced, using a microchip Mach−Zehnder interferometer (MZI), functional at specific wavelengths [91]. Inside the MZI system, light radiation travels across a ridge waveguide, furnishing one modal conduct. The light radiation current is split by a proportional Y-junction into two light streams, identical in all properties, and then reunited by a second Y-junction structure into the original light stream [92,93].

MZI systems were micromachined using reactive ion etching and utilizing 6-μm GaAs waveguide strips placed over the coating that resides on the GaAs underlying layer. The diameter of the MIR-MZI system was studied at 5 μm, in accordance with finite element methods, to confirm a single modal conduct. The implementation of the established MIR-MZI was illustrated by placing droplets at several spots across the MZI system, resulting in various intervention designs, as shown in Figure 5.

Depending on the deposition of water drops at different spots on the MRI-MZI system, different interferometric reactions can be anticipated, as shown in Figure 6. These kinds of tools may be exploited as a precise transducer technology for examining macromolecular structures and their configurational changes in MIR technology.

At present, there is much research being undertaken on integrated MIR-MZI instruments having a suitable microfluidic configuration, and also on the incorporation of optical components established on a waveguide implanted framework for increasing the performance of photons discharged by quantum cascade laser (QCL)-based and inter-band cascade laser (ICLs)-based instruments.

### 4.4. Silicon-On-Sapphire (SOS) Waveguides

There are some important material-based approaches for waveguide sensors functional in the 300 nm to 5000 nm region of the electromagnetic spectrum [94]. These methods, including silicon-on-sapphire (SOS), exploit the benefits of complementary metal-oxide-semiconductor (CMOS) technology. The advantages of SOS [95,96,97] and silicon-on-insulator (SOI) [98,99,100,101] in waveguide technology are determined by the MIR region of λ = 5.6 μm. Single-crystal silicon has optical clarity in the MIR region of λ = 8 μm and has a large refractive index of 3.4–3.5, whereas sapphire and silicon dioxide (SiO_2_) show wavelengths down to 6 and 3.7 μm, respectively. To resolve this issue, air-clad silicon waveguides have been suggested [91,95]. The intricate production techniques, e.g., disparate thin-film development, and complicated engraving methods have slowed the rate of development and prevented extensive operations. Si_3_N_4_ has optical transparency in the MIR region of λ = 8 μm and a refractive index of 2 [92], which is determined by the stoichiometry of its elemental composition. It has properties such as high chemical resistance and non-toxicity [93]. Silica is most commonly used under the layer utilized for Si_3_N_4_ operations.

To enhance the spectroscopic range up to 15 μm and obtain advantages from CMOS-based technology, and due to their higher refractive index, materials such as Si, Ge, GeSn, Si_3_N_4_, and aluminum nitride (AlN) are potential candidates for the fabrication of thin-film waveguides. These classes of materials have tremendous significance for waveguide fabrication technology, due to their higher MIR optical clarity, chemical inertness, and adaptability.

### 4.5. Diamond Waveguides

Diamond has numerous remarkable properties, and the possibility of fabricating excellent grade synthetic diamond has paved the way for the production of diamond-based devices. Apart from the exceptional physical and chemical properties of diamond, its optical features are very exciting, due to its wide transparency in the MIR spectroscopic region. By integrating nickel–nitrogen elements into the diamond chemical structure, the efficiency of a single photon source is improved, and this is highly desirable in optoelectronics and for quantum key distribution (QKD) [102,103]. Moreover, progress in diamond-based waveguides is essential for monolithic programs that exploit technologies such as quantum computing and QKD [104]. Diamond has a very important place in various analytic operations, such as atomic force microscopy (AFM), chemical sciences, and attenuated total reflection (ATR), in conjunction with infrared spectroscopy (IR-ATR spectroscopy). Moreover, diamond is distinguished by a low thermal expansion, chemical inertness, and exceptional toughness; additionally, by changing the surface chemistry of diamond, it could be utilized in advanced sensing applications. For use in chemical/biological sensing systems, diamond is a potential material for waveguide-based lab-on-a-chip structures.

Even though diamond has these amazing features, its implementation for waveguide geometries is difficult, due to the presence of monocrystalline films of millimeter sizes that are not suitable for monolithic optical integrated circuits [105]. Many crystalline diamond films have defects in their crystal lattice. Such crystallographic defects in polycrystalline diamond films may be due to lattice irregularities; therefore, they are not applicable for planar waveguides. The most important method for reducing these crystallographic defects is to develop films of larger proportions, because the lattice imperfections decrease as the film thickness increases, and this can result in free-standing optical waveguide films [106]. However, the resultant films may experience a thermal strain that influences their mechanical strength [107,108].

### 4.6. Chalcogenides

Chalcogenides are another class of outstanding materials for thin-film-based MIR waveguides. Chalcogenides consist of the materials S, Se, As, and Te, which are famous for their wide optical regime and higher refractive index, as in fiber optic applications. By manipulating the material configuration and ratio various thin-film waveguides can be constructed, due to the huge difference in refractive index between the substrate and chalcogenide surface. By utilizing GeTe_4_ as a waveguide basic layer having a refractive index of 3.34 and a ZnSe substrate with refractive index of 2.43, an appropriate absorbance difference can be produced between the basic layer and substrate, therefore enhancing the magnitude of the evanescent waves and allowing a higher surface responsiveness at the waveguide and sample intersection. In addition, by exploiting such material systems, channel frameworks for waveguides have been accomplished in the 2.5–7.5 mm spectral window [109]. One research team reported a manufacturing technique for these waveguide systems utilizing sputtering and lift-off methods to accomplish a GeTe_4_ waveguide layer on ZnSe substrates, which was subsequently adapted in sensing applications (Figure 7).

These third-generation waveguides are amorphous and yield a MIR window in the spectral region of 2–20 mm. Even though a single-mode nature is aspired to for many sensing investigations, a GeTe_4_ waveguide layer with channels can exhibit a multimode behavior for specific wavelength regions. Subsequently, these materials and manufacturing methods could be employed to obtain single-mode conduct [109]. The combination of third-generation channel waveguides and MIR fiber optics as integrating components suggests that these platforms are not absolute, but may work in synergy, depending on the specific applications.

### 4.7. Germanium Waveguides

Germanium (Ge) is one of the most extensively used chemical elements in MIR spectroscopic analysis and is normally utilized as monocrystalline waveguide molded into ATR components, due to its wide pellucidity across the suitable MIR spectroscopic window. Ge substrates are widely adaptable to the silicon operating procedure and also have the benefit of biostability. Nevertheless, the production of lab-on-chip third generation Ge waveguides remains difficult, because of the substantial dispersion drop in the Ge framework. A Ge layer placed on a ZnS substrate was produced by shining Ge prisms with 2 mm × 30 μm dimensions, therefore displaying an enhancement in surface reactivity [110]. The research team of Herzig and co-workers introduced single crystalline lab-on-chip Ge waveguides on a silicon layer for MIR spectroscopic techniques. A 2-μm wide single crystalline Ge film was developed using reduced-pressure chemical vapor deposition (RP-CVD) on a silicon support, which was shielded by an underlying silicon barrier layer, to decrease its absorption properties [111]. A single crystalline Ge film has excellent optical properties, consequently promoting the fabrication of Ge waveguides. Photolithography has been exploited to determine the waveguide patterns over Ge films. Subsequently, the patterns were shifted on the wafer via reactive ion etching (RIE), to obtain exceptional grade waveguides. To achieve exterior sides with better optical features, smoothing was performed to obtain better light radiation interactions (Figure 8). The waveguides were constructed with dimensions of 15 μm × 500 μm, connecting a fragment, a long funnel-like segment, and an elongated waveguide fragment. This waveguide fragment was devised to act in uni-mode with dimensions of 2.9 μm × 2 μm. To prevent random light interactions during experiments, a bending structure perpendicular to the fiber axis was introduced to allow the analysis of samples. Subsequently, a microfluidic structure was combined with a quick paradigm exploiting NOA81, a liquid adhesive (Figure 8) [112]. NOA81 exhibits an enhanced chemical stability and optical transparency. The microfluidic system exhibited excellent adherence following additional curing and heating operations. The optical arrangement, composed of a QCL functioning as an MIR light emission point, was utilized in single mode form. An IR light source was integrated into the waveguide structure utilizing ZnSe lenses, and the reaction between the evanescent waves and specimen took place in the microfluidic section. Cocaine in tetrachloroethylene (TeCE) was chosen as a model specimen at QCL wavelength as the transmission area of TeCE lies within the MIR spectroscopic region. Figure 8 shows an estimation of the optical radiation at outer surface of the Ge waveguide structure as the cocaine concentration was regulated. Therefore, the integration of a Ge waveguide platform and the microfluidic technique, together with the possibility of on-chip MIR observation of cocaine, was achieved [113].

### 4.8. HgCdTe Waveguides

Mercury-cadmium-telluride (MCT) sensors are extensively used photodetector materials because of their high sensitivity in the MIR spectroscopic range. MCT is considered an interesting material for the fabrication of MCT waveguides. Mizaikoff and his research team were the first to report the practical application of MCT waveguide systems, acting as excellent sensors for MIR detection platforms. An epitaxially technique was utilized to develop thin-film MCT films on a cadmium-telluride underlayer. MCT waveguides were developed exploiting MBE on a CdZnTe substrate, and having a refractive index of n = 4.0. Subsequently, the MCT film was split across its central axis, to acquire a chip of length 10 mm and width 8 mm, furnishing side walls with reasonable features for effective optic-integration and with minimal dispersion losses in the MIR spectroscopic region. Subsequently, split waveguide chips were placed in an optical structure consisting of an adjustable QCL light origin, ZnSe optical lenses for precise regulation of the laser emission, and a liquid-nitrogen equipped sensor. To evaluate the analytic ability, drops of mixture of 2-propanone and 2-propanol were placed at the MCT waveguide facet. Estimation at 1710 cm^−1^ detected the distinguishing absorption band of 2-propanone. Reduction of the light radiation strength, due to evanescent waves being influenced by the specimen under study, was examined at different 2-propanone ratios and provided a linear association. In relation to earlier reports utilizing GaAs/AlGaAs third-generation waveguides, the results of these MCT thin-film waveguides were slightly better. An increase in responsiveness could be achieved by microfabricating MCT waveguides utilizing the dry etching technique and thereby resulting in sideways detention, as previously reported in GaAs waveguides. Monolithically incorporating the waveguide structure and detector has tremendous potential for bio-chemical sensing platforms and lab-on-a-chip structures.

### 4.9. Silver Halides

The first MIR-based sensing platform for aqueous medium samples utilizing planar silver halide (AgC_l0.4_Br_0.6_) waveguides was reported in 2005 [114]. Silver halide films of dimension 190 μm × 3 mm were grown on tubular fiber fragments, utilizing a tapering technique to fabricate multiple-mode waveguides. Quantum cascade lasers (QCL) with radiation frequencies at 1650 cm^−1^ and 974 cm^−1^ were studied, to identify acid amide peaks in urea particles and the CH_3_−C absorption characteristics of ethanoic anhydride dispersed in methyl cyanide. Transmission estimations were performed utilizing an MCT sensor for the samples placed on the waveguides, and these were matched with the commercially available FTIR spectrum, where light from the source was incident on the surface of the waveguide and the readout signal was analyzed by the MCT sensor. After depositing a 0.01 mL sample of urea on the waveguide surface, a limit of detection (LOD) of 0.0807 mg was obtained, which was much better in comparison to a FTIR equipped with a waveguide structure. Similarly, under the same set of conditions, ethanoic anhydride was tested, and a LOD value of 0.0108 mg was obtained, which was comparable to utilizing the FTIR waveguide structure. The main cause of acquiring an identical LOD value for ethanoic anhydride was the overlapping of its frequency and the side mode of ethanoic anhydride. Another research group also reported the manufacturing of grating couplers utilizing an ionic beam on a silver halide waveguide, to provide an efficient coupling waveguide system [115]. QCL read-out reduction was reported, by measuring the intensity of a laser pulse with different ratios of ethanoic acid, and a standard curve was created. Halides have a broad spectral window in MIR and are not appropriate for hydrated specimens, due to their instability and absorptive nature. Thin-film halides are very complicated to develop. Selected waveguide material for analyte detection is shown in Table 1. 

### 4.10. Advantages

By exploiting the benefits of IR signal enhancement methods through surface-enhanced infrared absorption (SEIRA), spectroscopy allows biomolecular observation even in complicated biological structures and at lower volumes. For instance, transmembrane proteins have been functionalized on an Au-ATR waveguide surface for investigating protein interaction kinetics. Therefore, examining cells or whole tissues with molecular information is absolutely feasible [121]. Mizaikoff and collaborators recently demonstrated transepithelial liquid transfer along lung epithelial cells using a deuterium oxide dilution approach, established on such platforms [122]. Around a 0.06% vol/vol change was accomplished with this approach, which allowed the observation of changes of less than 24 nL, with the absence of any need for labeling. Therefore, this approach is most suitable for quick and varied cell physiological liquid transport research of live samples, as is essential in biomedical and pharmaceutical applications.

### 4.11. Disease Diagnostics with MIR Thin-Film Waveguides

At present, clinical diagnostic methods, including flow cytometry and enzyme-linked immunosorbent assay (ELISA), utilize a fluorescence detection technique that needs highly-trained medical professionals to label biological molecules and carry out assay protocols and that can need many hours to achieve results. On the other hand, optical biosensor platforms are label free and yield an immediate response [123]. Moreover, this platform only needs a small amount of reagent, which is beneficial, as they can be very costly (e.g., antibodies). On-the-spot observation is very attractive if the kinetics of a biospecific pair are needed, as in pharma industries. An optical biosensor platform is influenced by surface plasmon resonance (SPR) biosensors [124,125]. Regardless of the accomplishments of SPR biosensors, their price, miniaturization, and multifaceted use are important issues. Metal and dielectric resonators and waveguides have been introduced to address these issues [123]. For example, long-range surface plasmon polaritons (LRSPPs) can be used, in which transverse magnetic (TM) surface waves propagated on a thin metal pattern arrayed by dielectrics of the same refractive index [126].

Urinary tract infection (UTI): The conventional method for disease diagnosis of UTI is the identification of a pathogen in the clinical environment. The most effective method to observe and analyze bacterial pathogens is in patient urine [127]. This procedure has a good specificity for bacteria and a remarkable sensitivity, but requires a lot of time in a pathology laboratory. LRSPP straight waveguide platforms were utilized to selectively identify Gram-positive and Gram-negative pathogens in a human urine culture with a low sample concentration [128]. These investigations were performed utilizing filtered urine loaded with predetermined concentration of pathogens. Waveguides were functionalized with Protein G and antibody against Gram-negative pathogens. Figure 9 shows the observed optical power when four filtered urine test samples were injected; first, without bacteria (Urine1203); second, with a low concentration of Gram-negative bacteria (Bact1202B, e.coli); third, with a high concentration of Gram-positive bacteria (SEPI1126, s.epi); and fourth, with a low concentration of Gram-negative bacteria (Bact1202A, e.coli) [128,129]. It is essential to end the flow, so that the pathogens can settle down on the waveguide surface.

Leukemia: At present there is no well-established approach for the prompt detection of leukemia. Moreover, contemporary diagnostic tests normally involve complicated methods, including fluorescence-activated cell sorting (FACS), blood cell morphology, and medulla ossium biopsy. B-cell leukemia is distinguished by a high tumor cell count and, therefore, an unusual dispensation of G kappa (IgGκ) or lambda (IgGλ) in serum [131,132,133]. An investigation in leukemic patient blood was carried out by examining IgGκ-to-IgGλ ratios (κ: λ) in serum, utilizing an LRSPP straight waveguide platform. An Au waveguide surface was optimized with Protein G, using quick physisorption on Au, and two immobilizing methods were examined: (1) a novel reverse approach, in which the Protein G surface was optimized with patient blood serum; and (2) a classical direct method, where the Protein G surface was optimized with goat anti-human IgGs, then examined against patient blood serum.

Dengue: Dengue is a viral infection transmitted to human beings through the bite of infected mosquitoes, with approximately 390 million cases of infection annually [134]. It has been reported that more than half of the world’s population is at risk from infection, and this needs immediate attention [135]. At present, laboratory detection methods for dengue infection are costly, time-consuming, and require well-trained medical professionals [136]. Due to the lack of a specific treatment and an efficient vaccine, prompt diagnosis of dengue is essential to decrease the death rate of this infectious disease. Virus constituents that includes RNA and NS1 antigen can be detected in infected blood at the commencement of symptoms and, subsequently, by the growth of virus-specific IgM and IgG after a few days [137,138,139]. LRSPP straight waveguide platforms were employed to investigate infected blood plasma for specific IgM antibody and specific NS1 antigen [140,141]. Subsequently, a waveguide surface was developed with a carboxylated self-assembled monolayer, and the sample was optimized onto the Au surface utilizing carbodiimide chemistry. A plasma-functionalized Au surface was found to be efficient for reducing the nonspecific binding. As shown in Figure 10, two dengue detection approaches could be carried out by introducing separate biological recognition elements over a plasma functionalized surface; anti-NS1 monoclonal antibodies (MAbs) can be utilized for the detection of NS1 antigen, and dengue virus (e.g., serotype 2) for the detection of IgM antibody.

## 5. MIR Established Lab-On-A-Chip

Lab-on-chip technologies use these platforms, where various laboratory activities can be performed at miniaturized scale, and which includes chemical synthesis and analysis on a single chip, resulting in the formation of portable devices. In particular, a LOC is a devise that has the ability to perform laboratory operations at the scale of chip-configuration. The dimensions of the LOC chip may be around a few millimeters to a few centimeters [142].

LOC is actually the fusion of different technologies, which include microfluidics, electronics, optics, and sensing platforms [143]. LOC technology is a very effective approach for the early detection and diagnosis of life-threatening diseases. The main rationale of LOC-based platforms is the on-the-spot requirement for state-of-the-art pathological examination. The introduction of modern technologies that include MEMS and NEMS, and the incorporation of different interdisciplinary platforms on a single chip are viable, as shown in Figure 11.

Figure 12 displays the series of steps contributing to the lab-on-chip manufacturing process. The LOC process starts by withdrawing the biological sample, and this is followed by obtaining a specific analyte/biomarker. The next step involves the transducer, to act on the analyte/biomarker electro-optically, and this is determined by the biomedical application under investigation. This step is followed by energy conversion and amplification of the transducer signal, and this depends on the chosen application. Eventually, the amplified signal is processed utilizing microelectronics technology for data detection and analysis. Recent data suggests that this is a hot research area, and several research groups from well-reputed universities have devoted their time and efforts to this research area. Their purpose is to study and understand naonofluidics and biosensing, to integrate micro/nano engineering with the biological fields, to establish advanced technologies for LOC, and to evaluate new LOC applications. A LOC is a tool that has the ability to perform single or multiple laboratory operations in a chip format. A very small quantity of biological fluid, volumes of less than picoliters, is required by LOC technology. By exploiting LOC technology, high scalability and automation are achievable. LOC tools are usually designated “μTAS” (micro total analysis systems) and are devices that automate and incorporate all important processes for the chemical investigation of a biological sample (such as, sample collection, sample transport, chemical reactions, and detection), and are also a subgroup of MEMS (micro-electromechanical systems). An LOC platform is connected to microfluidic technology and involves the investigation and exploitation of a small volume of bio-fluid. An μTAS usually demonstrates the assimilation of a series of steps involved in the lab processes to carry out chemical investigation; on the contrary, a LOC platform is devoted to the assimilation of a few of the lab processes in a single chip. LOC based platforms have great significance in the medical field, including for the diagnosis of HIV infections and in the area of phytology.

MIR-established chemical and bio-sensors are an area of great interest. While chemical sensors based on this technology are very well supported by the literature and frequently employed, biosensing platforms established on MIR have been rarely documented. The principal reason for this being a less explored field is the inadequate sensitivity of MIR technology, and it is suitable for determining low loads of organic molecules e.g., amino acids, nucleic acids, proteins, RNAs, DNA, etc. The practicability of free DNA analysis using MIR-based biosensing techniques has been reported in the recent past [144]. There are platforms for considerably increasing the reactivity, without any extra efforts or the need to develop programs. Innovative surface alteration methods, decreasing of specimen quantities, and matrix effects are the key parameters constituting MIR-based biosensing platforms.

There are various methods/approaches available to enhance the sensitivity of MIR-based biosensing waveguides. The exploitation of applications such as surface-enhanced infrared absorption (SEIRA), and the introduction of some group 10 and group 11 metals, are known to increase IR absorption impressions. The synergetic effects of SEIRA and (Au) nanomaterials in a thin-film glass waveguide for modifying the absorption impressions of 4-nitrothiophenol was reported in the literature [145]. In addition to these methods, sensitivity can also be enhanced by chemical methods, e.g., use of next-generation molecularly selective materials and the enhancement of components using the evanescent waves of waveguide structures. The less intricate biomolecules of interest would require uncomplicated methods, e.g., well-established polyesters functioning as an enhancement layer and expediting the quantification of low quantity samples.

Volatile and semi-volatile organic constituents (VOCs and SVOCs) have been determined at μg/L (i.e., low ppb) levels in various aqueous matrices, including surface waters, aquifers, and even sea-water [146,147,148,149,150,151,152].

With the integration of suitable microfluidic structures on top of a waveguide, sophisticated extraction and enrichment strategies may be introduced, as recently shown for the detection of cocaine in human saliva [113]. Thereby, the analysis of real-world biomedical samples, such as blood, plasma, saliva, or urine is possible; providing on-chip or on-device sample preparation clearly benefits more complex IR-lab-on-a-chip LOC approaches.

A bibliometric investigation is one of the most convenient ways to understand a research area and the potential for research. An investigation produces better research output data, making it easy for scientists to evaluate the future developments of a particular topic. VOS viewer software was utilized to chart the data obtained from the Scopus database and organized by applying a wordlist cluster investigation [153]. Figure 13 depicts the LOC themes established, along with the number of articles published in this area, based upon data obtained from the Scopus database. In this configuration, the color-codes illustrate the number of identical zones and their associations attributed to a theme. The map has been divided into five clusters, depending on the wordlists used for LOC and their applications. For example, Cluster I (red color) consists of 378 published articles presenting the themes of protein synthesis, biosensing, and nanotechnology, using wordlists such as protein, nanofabrication, bioassay, and metal nanoparticles. Similarly, Cluster II (green color) consists of 255 published articles, presenting the themes of drug delivery and healthcare. The wordlist food safety and food analysis formed the theme food technology, and Cluster III (blue color) consists of 115 published articles. Additionally, the wordlist of genetic engineering formed the theme of Cluster IV (yellow color), comprised of 100 published items; and, lastly, pathology formed the theme of Cluster V (purple color), comprised of 115 published items [154].

### 5.1. Graphene as a Lab-On-Chip Material

In several research fields, carbon and its allotropic forms have great significance, due to their versatile physiochemical features, which enable them to be utilized in LOC devices for various operations. The lower quantities of sample required in LOC-based devices make them demanding and vulnerable to surface adsorption. The integration of carbon materials such as graphene into LOC devices is a good method to address this issue. Out of the various micro/nanomaterials present, graphene has gained significant interest for microfluidic operations, because of its excellent physicochemical properties, e.g., outstanding electrical and thermal conductivity and good surface–volume proportion. On top of these features, graphene possesses some special characteristics that render it very intriguing for microfluidic operations. The incorporation of graphene permits biofluids to pass on exterior side walls or through the internal central core [155]. This increases the prospects of controlling micro-sized biomolecules. Moreover, these minute biomolecules are suitable for the exploitation on dense arrays of graphene, in which the surface layer is in contact with the biofluid, decreasing the quantity and area substantially [156]. The incorporation of graphene has increased the biosensing ability of LOC devices to a great extent. Graphene-incorporated LOC devices have been successfully utilized for the identification of SARS-CoV-2, and this has captured the global attention of researchers to work in the same field [157].

The raw materials used for LOC devices should have a number of remarkable features, and materials such as graphene have potential for LOC device production [158,159]. The integration of graphene helps to accumulate more energy from the entire UV–vis spectrum. The classical honeycomb structure of light-weighted graphene has exceptional mechanical properties and pliability. Due to this mechanical strength, graphene is an excellent nanomaterial for the production of clinical LOC devices [160]. Graphene is regarded as the world’s most conductive material, with a conductivity of 2.12 × 10^5^ S/m [161,162]. This property of graphene helps in obtaining the electrochemical information in LOC devices. Graphene has an amazing specific surface area of 2630 m^2^/g, which is critical for the loading of biomolecules, to attain specific molecular information in LOC tools [163]. Graphene and its derivatives (e.g., graphene quantum dots) have been extensively exploited in LOC-based platforms [164].

### 5.2. Incorporation of Graphene into Waveguides

Researchers have suggested the integration of graphene with numerous photonic platforms, such as plasmonic systems [165,166], fiber optics [167], and photonic integrated circuits (PICs) [168,169], to improve the light–matter relationship in graphene-based photonic devices. The light–matter relationship can be improved by incorporating graphene with PICs, using the effect of the evanescent field of the optical waveguide. In relation with other arrangements, the production process of waveguide-incorporated graphene photonic devices is consistent with CMOS, due to which, the higher density incorporation of graphene is possible at a low price. Several waveguide-incorporated graphene photonic devices have been reported, which include an optical filter [3], saturable absorbers [170], modulators [171,172], and a light-sensitive detector [173]. Owing to these features, graphene-on-silicon PICs represent an attractive technology for the on-chip development of information and communications technology (ICT). Moreover, 5G telecommunication and internet over things (IoTs) require tens of billions of active and linked devices [174], which requires an operating speed of more than 25 Gb s^−1^ and a power consumption less than 1 pJ bit^−1^ for essential constituents [175], such as light-sensitive detectors and modulators. As reported in the literature, the international silicon photonics market was evaluated to be around USD 0.8 billion in 2018 and is likely to grow to USD 2.0 billion by 2023. Graphene’s excellent optical features may assist in improving the optoelectronic efficiency of the normal PICs, to satisfy the rapidly evolving ICT market demand. In addition, the incorporation of graphene in waveguides can enhance the waveguide adsorption ability of chemical species, which results in increasing the sensitivity of on-chip detectors. Hence, waveguide-incorporated graphene photonic devices pioneer the method of on-chip optical sensing platforms for advanced ICT.

### 5.3. Graphene as a Biosensor

An electrochemical biosensor (EBS) is a sensing tool that works by converting biochemical information into an electric signal. A typical EBS comprises two components: a surface related biocomponent that works with the elements of concern in the blood stream, and a transducer that works by converting the acquired information into a measurable signal.

Electrochemical techniques have gained great attention based on their attractive properties, such as being simple, less expensive, and portable, and for small scale detection of samples [176]. EBS has great significance, and graphene is a frequently utilized material as an EBS electrode. The amazing surface area and outstanding electrical conductivity of graphene and its allotropic forms allow efficient biomolecular adsorption and express electron transport through redox areas of the electrode surface, which helps with the precise identification of biomarkers. The best way to develop the electrical proximity between graphene and the electrode is to grow graphene directly on the electrode substrate, which can realize contamination-free biosensors. This direct and fast growth of graphene can produce nanographene on the surface of the electrode. Owing to the geometry of graphene, carbon atoms are situated on the surface, rendering graphene extremely sensitive to its surroundings. Hence, even slight variations in the surroundings can cause a substantial shift in their conductivities, which boosts its potential for sensing applications [163].

The careful identification of nucleic acids, which is very important for holding and transferring genetic data is helpful in biological analysis. Graphene incorporated EBS have been discussed in the literature for DNA identification with a super high reactivity and sensing window of 9.4 zM [177]. Similarly, a label-free EBS was suggested for high-speed identification of DNA, established on gold nanoparticles-toluidine blue incorporated graphene, with a sensing window of 2.95 pM [178]. Moreover, label-free highly sensitive identification of miRNA established on a EBS was also reported [179]. These graphene biosensors exhibited very good sensing ability for model miRNA, with effective sensing windows from 10 fM to 10 µM.

There are various graphene-established EBS being grown for precise identification of amino acids in cancer patients. A recyclable EBS established on a GO-designed Au electrode was reported in the literature. The GO was exploited as a potent transporter for bevacizumab drugs and allowed effective identification of the signaling protein VEGF in blood, with a sensing window of 31.25 pg/mL [180]. As shown in Figure 14, the research team designed the program by electrophoretically depositing reduced graphene oxide on a gold surface for amino acid identification, with a sensing window of 1 pM. By employing differential pulse voltammetry, a remarkable drop of current could be observed after completion of the process [181].

One research team reported exciting work, in which graphene-established amperometric biosensors were developed utilizing porous nanographene-designed electrodes, as shown in Figure 15. The fabrication of graphene was executed with the quick reduction of GO in the presence of Mg/Zn composite material. The resultant material exhibited increased reactivity in the presence of folic acid [182]. This peptide-folic acid-designed graphene EBS was developed for the identification of cervical cancer. The detection of cancerous cells was due to the precise interaction of the sensory-receptor and Vitamin B [183].

Although there have been many reports about graphene biosensors in the literature having reasonable stability and reproducibility, the results of some biosensors with bio-samples (e.g., urine, blood, etc.) are not satisfactory in their detection level, due to the non-compatibility of organic and inorganic molecules during graphene and biomarker interactions. One method to address this issue is to grow and establish ultrasensitive and large-specific-area biosensors [184].

## 6. Summary and Future Perspectives

MIR spectroscopic technology has implemented various microfluidic-based waveguides, e.g., GaAs/AlGaAs, (SOS), diamond, chalcogenides, Germanium, HgCdTe, silver halide waveguides, and biological analytes to establish fundamental principles, prior to the determination of bio-analytes for diagnostic applications. Researchers have achieved substantial progress with MIR waveguide materials and technology during the last few years. The evolution of areas such as light sources, sensitive detectors, and data analyzing software needed for the MIR spectroscopic techniques associated with biosensing applications, such as prompt detection of illness, has also progressed. This process has enhanced the possibility of acquiring MIR waveguide technology in various fields.

MIR spectroscopic techniques ensure the fast examination of various diseases, but currently this is only a laboratory procedure. Human biofluids are complicated and non-identical; therefore, an extensive database comprising biological and spectroscopic fingerprints needs to be established for precise identification of diseases. Top-quality estimation and benchmark data are of tremendous significance, and this needs immediate attention. Substantial and regular clinical investigations are needed before MIR spectroscopic techniques are developed into routine diagnostic tools.

Many LOC devices have been available on the market for years (e.g., home pregnancy tests, real-time glucose monitoring, etc.). Miniature portable LOC devices for point-of-care (POC) applications are being adapted for testing in economically developing countries, due to the inexistence of good healthcare and laboratories. These automatic, precise, and economical diagnostic tools do not require experienced medical professionals and can be used on-the-spot by patients. In LOC devices, which are classified as a subgroup of BioMEMS, electrical and mechanical parts are incorporated into the same arrangement. Generally, samples collected from patients (such as blood, saliva, urine etc.) are placed onto a single chip, and a series of steps such as separation, filtration, examination, and a read out of the information occurs on the miniature single chip in these platforms. The information received is displayed visibly, such as in pregnancy tests and blood glucose tests with a glucometer, and does not require the consultation of a medical professionals. Although LOC platforms have progressed in recent years, and are being increasingly utilized in practical clinical diagnosis, there are many significant challenges to be addressed. The development of advanced materials with porous and flexible membranes with better reproducibility is very important. Current LOC technologies regularly utilize some form of external assistance to carry out sample pretreatment prior to the detection method. However, to improve their potential for point-of-care testing applications in remote or undeveloped areas, and to increase their detection efficiency, these devices should be supported by effective on-chip and preconcentration platforms in future endeavors. Additionally, the reagents (e.g., enzymes, antigens, and antibodies) used in LOC devices must be able to withstand harsh environmental conditions in the course of shipment and storage, mainly in remote or undeveloped areas. The variation in sensitivity and selectivity of lab-on-paper devices may result in high levels of fake-negative and fake-positive results, which are also important issues to address. This requires active cooperation between researchers, medical professionals, and engineers to bring this technology to the point of care.

Absorption spectroscopic technology is a reliable tool that works in synergy with waveguide platforms to produce a set of data to investigate biofluids, for accurate results. Lab-on-chip devices involve the incorporation of optoelectronics and microfluidic technology on the same device, allowing a better strength and miniature size. It is anticipated that precise waveguide fabrication with controlled surface engineering will allow better estimation of biofluids in living environments. Graphene has many intriguing characteristics. Graphene can be integrated into microfluidic-based LOC devices, to modify their biosensing ability, biostability, compatibility, and surface functionalization, and also permits self-cleaning properties and maximizing the signal-to-noise ratio. The detection of the biomarker in biological samples using graphene-based biosensors is still in its early stages and needs immediate attention. As non-specific interconnections are of major interest in graphene-based intersections, the simultaneous absence of data and the mass-scale repeatability of the production of graphene-based biosensors are possibly the most critical factors that delay its commercialization. These issues need to be resolved to accomplish mass-scale manufacturing of graphene-based biosensors. Moreover, incorporating graphene with waveguides could solve many problems, such as their narrow bandwidth, higher energy utilization, and higher manufacturing cost than common tools. Due to the direct distribution of Dirac fermions, graphene can be assimilated with different substrates, to produce waveguides, sensors, detectors etc. The implementation of graphene waveguides can enlarge bandwidths from 300 nm to 6 μm or more.

Several challenges have yet to be addressed. The association between researchers with different backgrounds and expertise is of great importance to push this area forward and to promote graphene-based waveguides as an important diagnostic tool. The accomplishment of new graphene-based waveguides depends on their replicability and market value. The inception of various organizations producing mono- and multilayered graphene nanomaterials in many areas has been the inspiration for utilizing graphene in biosensing operations.

## Figures and Tables

**Figure 1 sensors-22-05443-f001:**
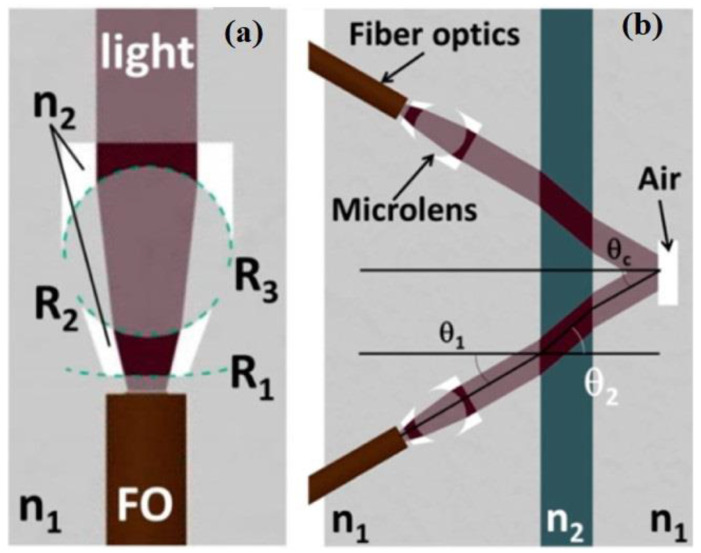
Graphic presentation of (**a**) the establishment and alignment of a collimating lenses and radius of curvature R_(1–3)_ developed for the utilization with fiber optics of a distinct numerical aperture, and (**b**) the arrangement of various micro-optic constituents to adjust the light–specimen relationship: Figure adapted from ref. [67] with permission of Nature Protocols. Copyright 2011 Nature Publishing Group.

**Figure 2 sensors-22-05443-f002:**
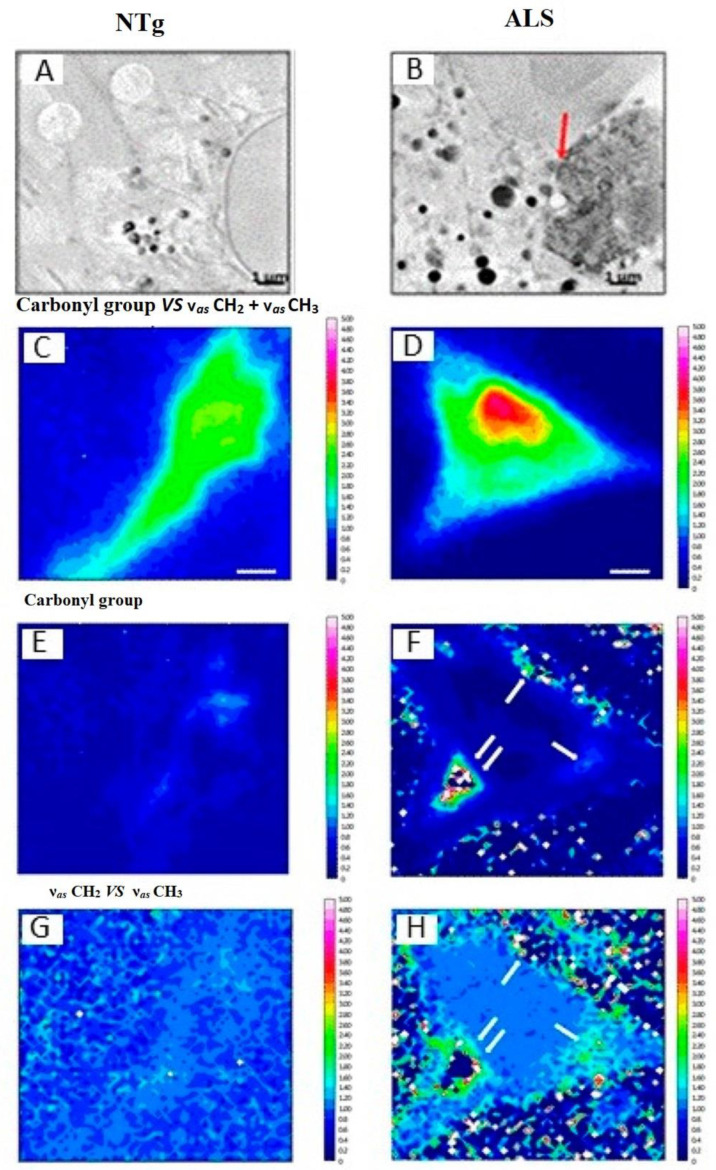
Synchrotron-FT-IR imaging of (**A**) Ntg and (**B**) ALS astrocytes, where (**C**,**D**) ν(C=O) vibrations, (**E**,**F**) ratio of ν(C=O)/(ν_as_CH_3_ + ν_as_CH_2_) vibrations, and (**G**,**H**) ratio of ν_as_CH_2_/ν_as_CH_3_ vibrations reveal concentrated intracellular lipid compositions and acyl chain unsaturation (white arrows) associated with lipid vesicles and lipid peroxidation. Reproduced with permission from ref. [76,77]. Copyright 2019 American Chemical Society.

**Figure 3 sensors-22-05443-f003:**
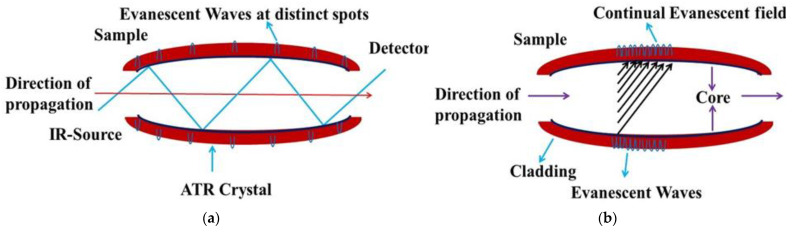
Graphical illustration of light propagation in an (**a**) ATR crystal and (**b**) optical waveguide.

**Figure 4 sensors-22-05443-f004:**
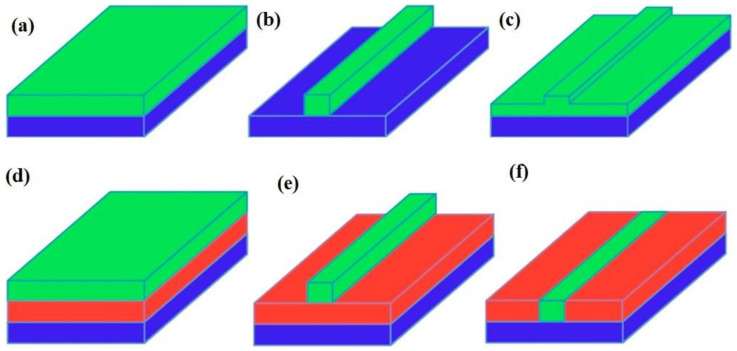
Illustration of common thin-film waveguide geometries: (**a**) slab waveguide, (**b**) strip waveguide, (**c**) rib waveguide, (**d**) slab waveguide, (**e**) ridge waveguide, and (**f**) embedded/buried waveguide. The same functional layers are marked with the same colors: green = waveguide layer (nc), red = optical buffer layer (nb), blue = substrate (ns). Reproduced with permission from ref. [90]. Copyright 2006 the American Chemical Society.

**Figure 5 sensors-22-05443-f005:**
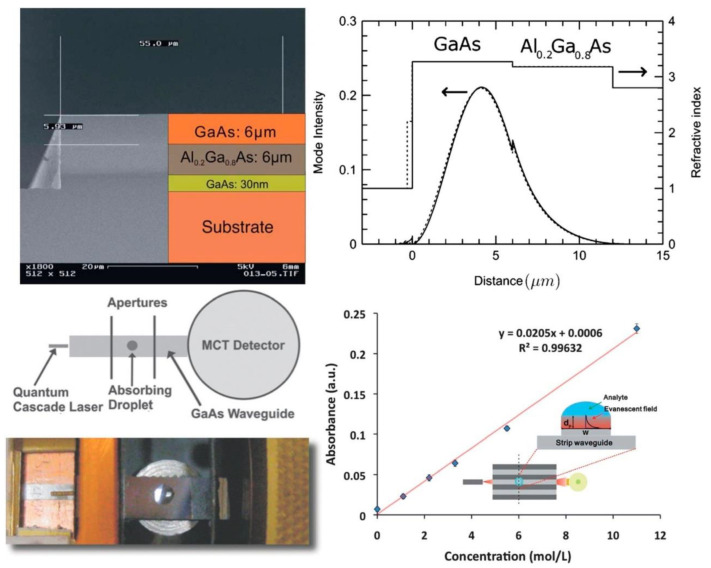
(**top left**) Cross-section of a MOVPE-grown GaAs/Al_0.2_Ga_0.8_As ridge waveguide obtained via reactive ion etching. (**top right**) Computed optical mode profile of the thin-film GaAs/AlGaAs waveguide structure with a layer dimension of 6 μm (left axis, mode intensity; right axis, refractive index). (**bottom left**) Experimental setup of a single wavelength emitting QCL pigtail-coupled to a GaAs slab waveguide. (**bottom right**) Analytical response to 2-nL droplets of an analyte in evanescent field absorption measurement with QCL pigtail-coupled to a structured GaAs ridge waveguide (200 μm in width). Reproduced with permission from ref. [12]. Copyright 2006 the American Chemical Society. Reproduced with permission from ref. [87]. Copyright 2012 the Royal Society of Chemistry.

**Figure 6 sensors-22-05443-f006:**
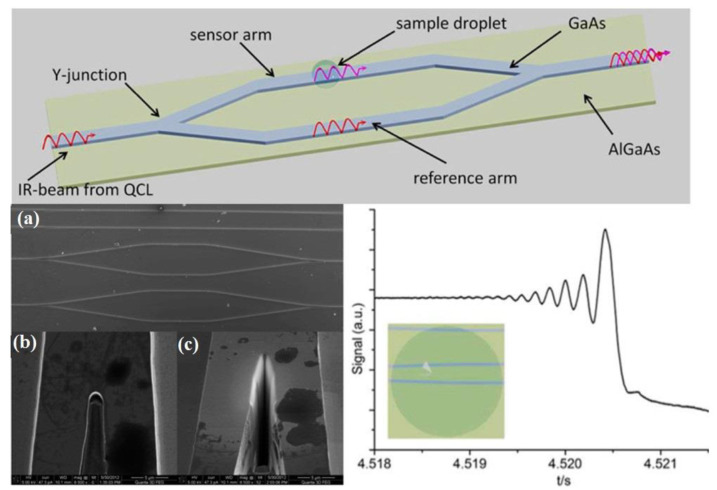
(**top**) Scheme of an on-chip MIR Mach−Zehnder interferometer. (**bottom left**) Scanning electron microscopy (SEM) images of MIR-MZI waveguides show (**a**) a top view of GaAs/AlGaAs MZIs, and the Y-junction (**b**) before and (**c**) after using focused ion beam (FIB) microscopy for refining the structure for the joint of the waveguide arms. (**bottom right**) Typical interferometric signal generated by depositing, e.g., water droplets at one of the MZI arms, resulting in a phase delay and giving rise to the observed interference pattern. Reproduced with permission from ref. [91]. Copyright 2013 the American Chemical Society.

**Figure 7 sensors-22-05443-f007:**
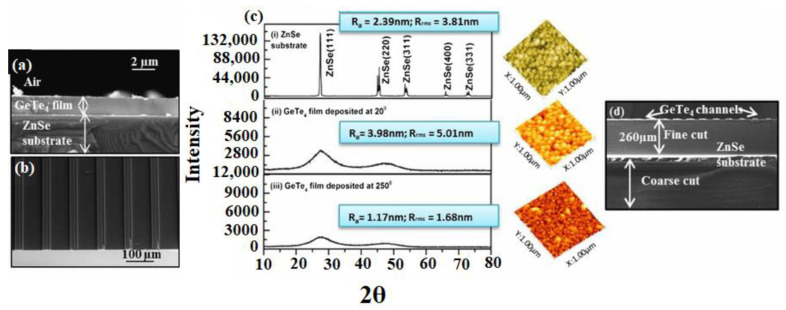
(**a**) SEM image of a cleaved cross-section of GeTe_4_ film as deposited; (**b**) top view of GeTe_4_ channels fabricated using lift-off techniques; (**c**) *X*-ray diffraction pattern and atomic force microscopy image of a ZnSe substrate and GeTe_4_ films. (**d**) Distal coupling facet of the waveguide cut by ductile dicing [109].Reproduced with permission from the Optical Society Publishing, 2015.

**Figure 8 sensors-22-05443-f008:**
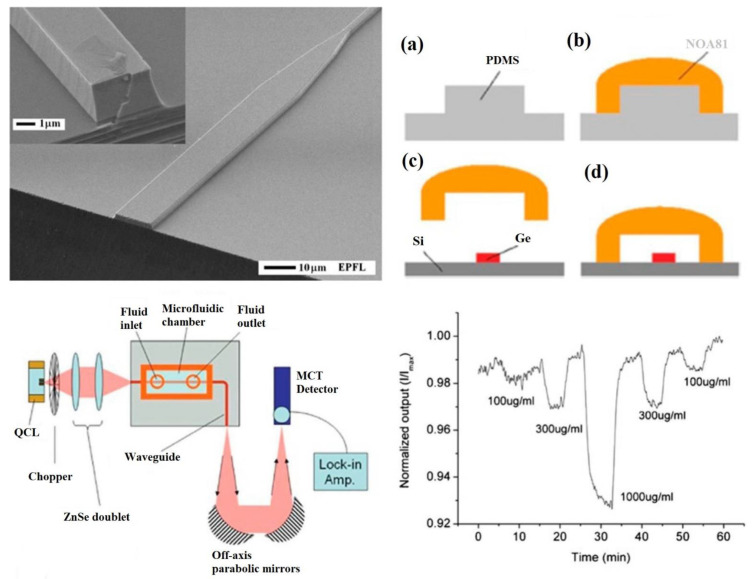
(**top left**) SEM image of a 2-μm monocrystalline germanium-on-silicon ridge waveguide. (**top right**) Fabrication scheme: a UV-curable adhesive (NOA81) cast onto a PDMS master and subsequently bonded onto the silicon substrate, forming a microfluidic channel, which was connected to a syringe pump. (**bottom left**) Optical setup: QCL radiation is coupled via lenses into the waveguide and via an off-axis parabolic mirror onto a MCT detector. (**bottom right**) Waveguide output during a dynamic analytical study using different cocaine concentrations present at the waveguide surface determined at a wavelength of 5.8 μm. Reproduced with permission from refs. [111] (Copyright 2012 Optical Society of America) and [112] (Copyright 2012 Royal Society of Chemistry).

**Figure 9 sensors-22-05443-f009:**
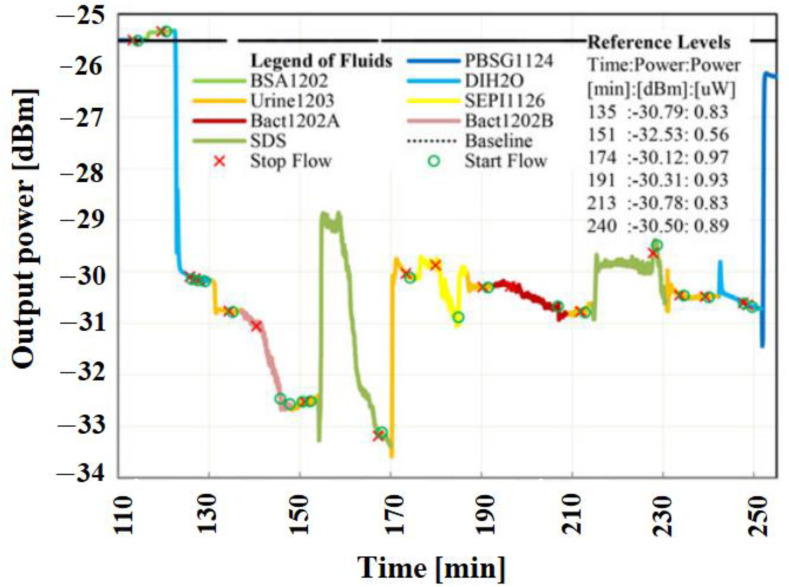
Selective detection of Gram-negative bacteria (*E. coli*) in urine using a LRSPP straight waveguide biosensor functionalized using Protein G and antibody against Gram-negative bacteria [128,129]. Bact1202A was a positive control urine solution with a high *E. coli* concentration of 10^10^ CFU/mL. Bact1202B was a positive control urine solution with a low *E. coli* concentration of 10^6^ CFU/mL. SEPI1126 was a negative control urine solution with a Gram-positive bacteria (s.epi) concentration of 10^12^ CFU/mL. The refractive index at 1310 nm of Urine1203 is 1.32276 and of PBSG1124 (buffer) is 1.33152. Copyright with permission from IEEE Ref. [130].

**Figure 10 sensors-22-05443-f010:**
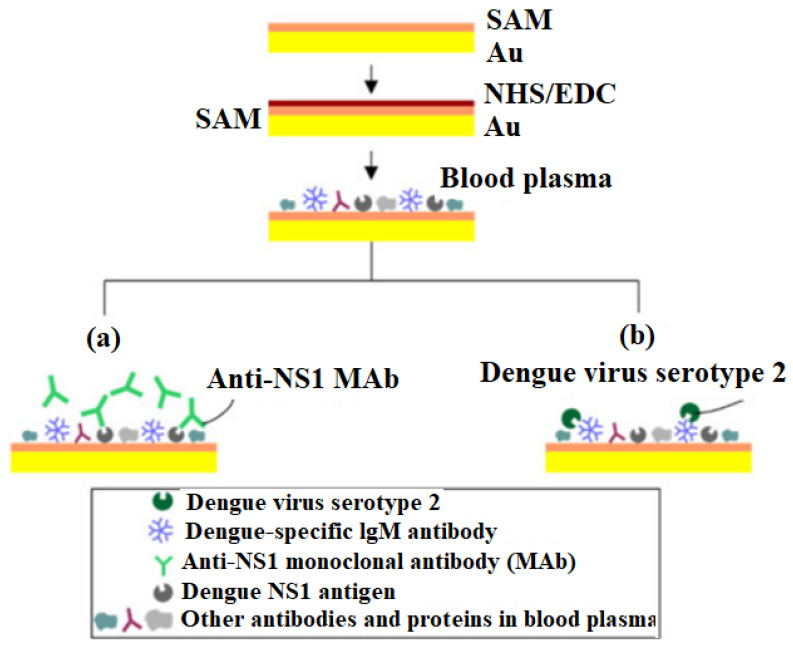
Schematic illustrating the functionalization of a waveguide surface for the detection of (**a**) dengue NS1 antigen and (**b**) dengue-specific IgM antibody. Copyright with permission from IEEE ref. [130].

**Figure 11 sensors-22-05443-f011:**
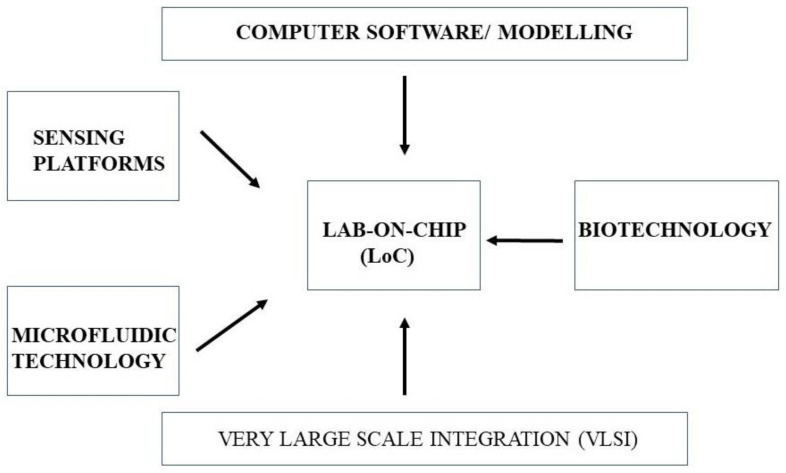
Interdisciplinary field of lab-on-chip technology.

**Figure 12 sensors-22-05443-f012:**
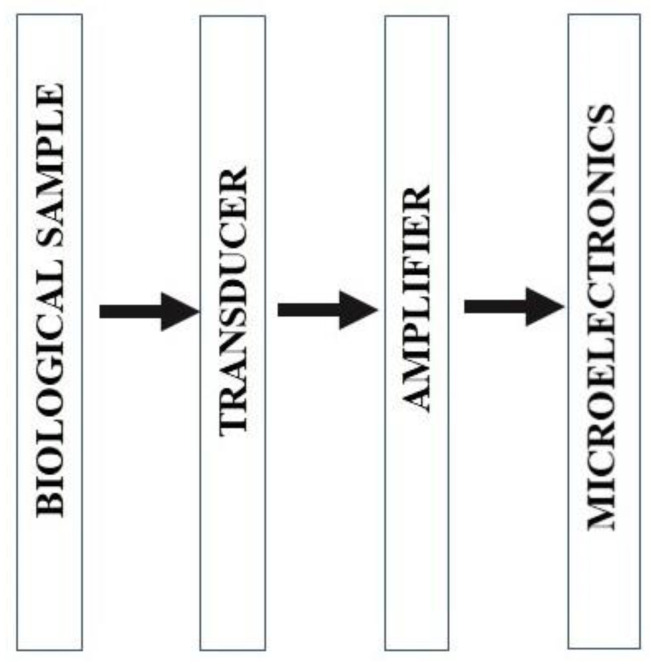
Schematic illustration of lab-on-chip technology.

**Figure 13 sensors-22-05443-f013:**
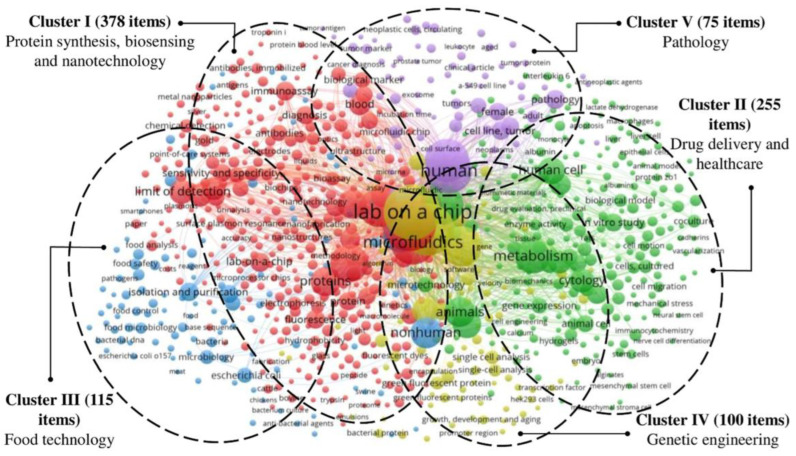
Lab-on-chip themes using a wordlist cluster investigation of the bibliometric data extracted from Scopus using VOS viewer software. Wordlist cluster investigation was performed to determine the research hotspots and developments in the last two decades. Copyright with permission from ref. [153] Springer.

**Figure 14 sensors-22-05443-f014:**
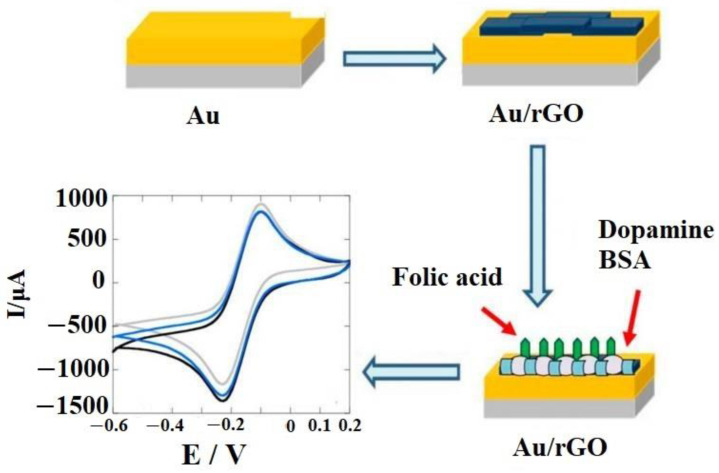
Electrochemical biosensor using a graphene-modified electrode for quantitative detection of folic acid protein. Reproduced with permission from [181]. Copyright 2016 Elsevier B.V.

**Figure 15 sensors-22-05443-f015:**
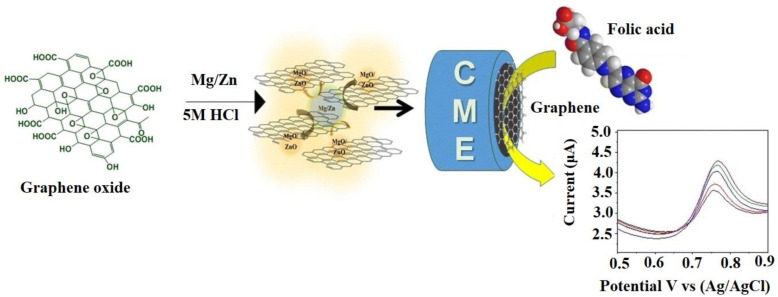
Graphene-based electrochemical biosensor for detection of folic acid. Reproduced with permission from [182]. Copyright 2016 Springer.

**Table 1 sensors-22-05443-t001:** Selected materials for analyte detection on a thin-film waveguide system in the MIR spectroscopic region.

Waveguide Material	Analyte and Volume Requirement	Refractive Index	Reference
GaAs	acetic anhydride (2 nL)	3.3	[17,18]
Si	D_2_O in water (35 μL)	3.426	[116]
Ge	cocaine in saliva	4.01	[113]
Si_3_N_4_	D_2_O in water (35 μL)	2.33	[117,118]
Al_2_O_3_ (Sapphire)		1.52	[118,119]
diamond	acetone in D_2_O (5 μL)	2.38	[120]

## Data Availability

Data underlying the results presented in this paper are available from the corresponding author upon reasonable request.

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
