# Peer review of "Advanced Waveguide Based LOC Biosensors: A Minireview"

_sensors, 2022, doi:10.3390/s22145443_

Round 1
Reviewer 1 Report
This mini-review paper (I think) gave a summary about Lab-on-chip devices for clinical diagnostics. The authors introduce this mainly based on MIR from several aspects, including fundamental principles, development stages, construction as well as spectroscopy techniques to microfluidic based LOC devices functionalized with graphene materials, covering 100 references.
To be frank, I am not high expert in this filed (I am only familiar with graphene part), but after reading this paper I also got more thoughts about LOC design which is beneficial for my future research. However, from my point of view, the structure of the contents is a bit messy (from my point of view) and not so easy to follow. Therefore, I recommend it for acceptance but after a major revision. The main suggestions are as below:
1. The title of this paper is about LOC Devices. It looked like the paper is going to talk about the devices based on LOC. However, the whole paper talked more about MIR from the very beginning without any links with LOC, and then talked about the principles, construction and so on mostly about MIR. The LOC was only mentioned somewhere in the paper. So, I think either the title or the contents should be modified.
2. So heavy texts in the paper, I suggest to summaries the works about LOC in a table or a drawing which would be helpful for readers to follow. Throughout the paper, more pictures from referenced papers are suggested to add.
3. The whole paper talked more, actually, the materials which can be used for LOC. This is beyond the aim of this paper.
4. In conclusion, the authors actually raised a lot of questions and focuses on graphene, but not much about LOC
5. The title is …..critical review, but from my point of view, it is more like a mini-review
6. It may be better to add a Content Index to help reader to follow.
Author Response
Reviewer 1
This mini-review paper (I think) gave a summary about Lab-on-chip devices for clinical diagnostics. The authors introduce this mainly based on MIR from several aspects, including fundamental principles, development stages, construction as well as spectroscopy techniques to microfluidic based LOC devices functionalized with graphene materials, covering 100 references.
To be frank, I am not high expert in this field (I am only familiar with graphene part), but after reading this paper I also got more thoughts about LOC design which is beneficial for my future research. However, from my point of view, the structure of the contents is a bit messy (from my point of view) and not so easy to follow. Therefore, I recommend it for acceptance but after a major revision. The main suggestions are as below:
Point 1. The title of this paper is about LOC Devices. It looked like the paper is going to talk about the devices based on LOC. However, the whole paper talked more about MIR from the very beginning without any links with LOC, and then talked about the principles, construction and so on mostly about MIR. The LOC was only mentioned somewhere in the paper. So, I think either the title or the contents should be modified.
Response 1: We would like to thank reviewer for giving us feedback on our manuscript. The title of the manuscript is modified in updated version. All changes have been done in purple bold phase. Also, the manuscript is well-structured following reviewers’ comments. LOC is highlighted in the revised manuscript wherever necessary. Also, two new figures (Figure 11 on page 39 and lines 1-13 and Figure 12 on page 40 lines 1-15, page 41 lines 1-10, page 43 lines 1-15, page 44 lines 1-7) on LOC are added to illustrate the concept.
Point 2. So heavy texts in the paper, I suggest to summaries the works about LOC in a table or a drawing which would be helpful for readers to follow. Throughout the paper, more pictures from referenced papers are suggested to add.
Response 2: The works about LOC has been mentioned in Figure 13 and Page 43 and lines 1-15, and new figures eg. (Fig. 2, on page 12 Fig. 9, on page 36 Fig. 10, on page 38 Fig. 11, on page 39 Fig. 12, on page 40 and Fig. 13 on page 43) has been added in revised version manuscript.
Point 3. The whole paper talked more, actually, the materials which can be used for LOC. This is beyond the aim of this paper.
Response 3: We have improved the manuscript. However, the materials we focused is basically selected for MIR wave guide applications (MIR active materials are heart and soul of LOC devices as active layer). Then the device fabrication and applications were also reviewed and summarized.
Point 4. In conclusion, the authors actually raised a lot of questions and focuses on graphene, but not much about LOC
Response 4: The conclusion has been modified and information about LOC has been added in page 50 lines 13-22 and 51 lines 1-14 in bold purple phase.
Point 5. The title is …..critical review, but from my point of view, it is more like a mini-review
Response 5: The title has been changed in revised manuscript.
Point 6. It may be better to add a Content Index to help reader to follow.
Response 6: Content Index has been added page 2 and page 3.
Please find attachment

Reviewer 2 Report
Kanjwal and Al Ghaferi present a description of spectrophotometric methods, an excursus on micro-optic components and IR spectroscopy, a discussion on the evolution of different waveguides and some examples of their use as biosensors as well as graphene-based biosensors. The overall structure of the paper is quite confused and the content does not reflect what is claimed in the title, which is indeed misleading. The title announces a critical review of analytical devices for clinical diagnostics, but in the introduction at lane 36 it is stated that the focal point of the manuscript is the examination of bio-fluids on thin film waveguides, which are a sub-class of analytical LOC devices. Therefore, the title should be changed to fit the content of the manuscript.
As mentioned, the manuscript suffers from various defects, the main relies in the organization of chapters. The introduction describes the aim of this review until lane 59, than all the rest of introduction is a sort of summary of part of the paper, but is not harmonized neither related with the previous part. Introduction should be rewritten to better cover the topic.
The second chapter explains the principles of spectroscopy, like in a lesson, but it is inappropriate, since they are universally known even to all students of the field. Section 2 and 2.1 should recall these principles if functional to the rest of the paper in a fused paragraph, which could comprehend also the paragraph 2.2.
Paragraph 2.3 should not appear in this position or should not appear at all. It is not clear how the method described in this paragraph, could be correlated with analytical lab-on-a-chip? Moreover, references should be cited and not only the firs author (see line 163, 173, etc.).
Chapter 3 is named Discussion, but describes MI waveguides as well as chapter 4 describes thin-film waveguides. Chapter 5 introduces LOC and their applications in general terms and not necessary linked to clinical diagnostics. Here, graphene appears for biosensing, but not related to waveguides.
The overall structure of the review should be rethought. A precise topic should be identified (e.g. biosensors based on waveguides, or LOC for clinical diagnostics, or applications of spectroscopic methods or…) and recent literature on this topic should be revised in detail, discussing criticisms and progresses reported.
Author Response
Reviewer 2
Point 1. Kanjwal and Al Ghaferi present a description of spectrophotometric methods, an excursus on micro-optic components and IR spectroscopy, a discussion on the evolution of different waveguides and some examples of their use as biosensors as well as graphene-based biosensors. The overall structure of the paper is quite confused and the content does not reflect what is claimed in the title, which is indeed misleading. The title announces a critical review of analytical devices for clinical diagnostics, but in the introduction at lane 36 it is stated that the focal point of the manuscript is the examination of bio-fluids on thin film waveguides, which are a sub-class of analytical LOC devices. Therefore, the title should be changed to fit the content of the manuscript.
Response 1: We would like to thank reviewer for giving us feedback on our manuscript. The title of the manuscript is modified in updated version. All changes have been done in purple bold phase. Also, the manuscript is well-structured following reviewers.
Point 2. As mentioned, the manuscript suffers from various defects, the main relies in the organization of chapters. The introduction describes the aim of this review until lane 59, then all the rest of introduction is a sort of summary of part of the paper, but is not harmonized neither related with the previous part. Introduction should be rewritten to better cover the topic.
Response 2: The chapters of the manuscript have been reorganized and the introduction has been updated and contents about mid-infrared spectroscopy and waveguides has been added in introduction.
Point 3. The second chapter explains the principles of spectroscopy, like in a lesson, but it is inappropriate, since they are universally known even to all students of the field. Section 2 and 2.1 should recall these principles if functional to the rest of the paper in a fused paragraph, which could comprehend also the paragraph 2.2.
Response 3: Now in the revised manuscript, section 2, 2.1 and 2.2 are condensed to give a brief introduction to common readers.
Point 4. Paragraph 2.3 should not appear in this position or should not appear at all. It is not clear how the method described in this paragraph, could be correlated with analytical lab-on-a-chip? Moreover, references should be cited and not only the first author (see line 163, 173, etc.).
Response 4: This section is included to give readers an idea about importance of infrared spectroscopy in clinical discovery and the references are properly cited (page 13 and line 8) in revised manuscript.
Point 5. Chapter 3 is named Discussion, but describes MI waveguides as well as chapter 4 describes thin-film waveguides. Chapter 5 introduces LOC and their applications in general terms and not necessary linked to clinical diagnostics. Here, graphene appears for biosensing, but not related to waveguides.
Response 5: The applications of disease diagnostics for the MIR thin-film waveguide has been added (page 34 lines 13-23 and pages 35, 36, 37, 38) and supported by Fig. 9 and Fig. 10. The role of graphene in waveguides is highlighted in the revised manuscript following reviewers’ suggestions. The incorporation of graphene into waveguides has been added, in the revised manuscript (page 45 line 14 to 23 and page 46 line 1 to 14).
Point 6. The overall structure of the review should be rethought. A precise topic should be identified (e.g. biosensors based on waveguides, or LOC for clinical diagnostics, or applications of spectroscopic methods or…) and recent literature on this topic should be revised in detail, discussing criticisms and progresses reported.
Response 6: The structure of the manuscript has been modified based on 85 more references has been added. Title has been changed. Some sections are condensed (section 2, 2.1 and 2.2) and some new sections are included (4.10, 4.11, 5.2) and some sections have been modified (section 1, 2.1, 5) and conclusion has been modified in revised manuscript.
Please find attachment

Reviewer 3 Report
This article describes the MIR spectral waveguide lab-on-chip devices for clinical diagnosis, advanced waveguide materials, the benefits and future of waveguide-established MIR spectroscopy for diagnosis, incorporation of graphene into waveguides improving the light-graphene interaction, etc. It is meaningful and interesting, but some points need to be further clarified before publishing.
1. Some places (like lines 32-34, 113-114,164-167, etc. ) of the manuscript need to add references.
2. In the introduction section, the conceptual content of mid-infrared spectroscopy and waveguides are not given enough explanation.
3. Please elaborate on the advantages of the thin film waveguide platform for MIR spectroscopy. Whether it can overcome the effect of the liquid present in the MIR spectrum on the sample?
4. It lacks an overview of the overall content framework of the manuscript in the introduction section.
5. The logic of some parts of the manuscript is a bit confusing, for example, the content of section 2.3 does not match the title of chapter 2(Fundamental Principles: Spectrophotometric Evaluation on Chip).
6. Some applications of disease diagnostics for the MIR thin-film waveguide need to be supplemented in the description of waveguide materials. (chapter 4 of the manuscript)
7. In section 5, the incorporation of graphene into waveguides is not discussed thoroughly. Please add the relevant content.
Author Response
Reviewer 3
This article describes the MIR spectral waveguide lab-on-chip devices for clinical diagnosis, advanced waveguide materials, the benefits and future of waveguide-established MIR spectroscopy for diagnosis, incorporation of graphene into waveguides improving the light-graphene interaction, etc. It is meaningful and interesting, but some points need to be further clarified before publishing.
Point 1. Some places (like lines 32-34, 113-114,164-167, etc.) of the manuscript need to add references.
Response 1: We would like to thank reviewer for appreciating our work. We have included all critical suggestions of the reviewer which improved overall quality of our manuscript. All changes have been made in purple bold phase. The references no (ref. no 2,3, on page 3 line 19 and ref. no 66, 67, on page 9 and line 7 and ref. no 78, 79 on page 13 line 8) has been added in revised manuscript.
Point 2. In the introduction section, the conceptual content of mid-infrared spectroscopy and waveguides are not given enough explanation.
Response 2: conceptual content of mid-infrared spectroscopy (page 4 lines 3-23 and page 5 lines 1-13) and about waveguides (page 6, line 1-2, page 7 lines 1-23 and page 8 lines 1-4) have been added in revised manuscript.
Point 3. Please elaborate on the advantages of the thin film waveguide platform for MIR spectroscopy. Whether it can overcome the effect of the liquid present in the MIR spectrum on the sample?
Response 3: This information has been updated in revised version (page 34 and lines 1-12).
Point 4. It lacks an overview of the overall content framework of the manuscript in the introduction section.
Response 4: The introduction has been improved following reviewers’ suggestions and contents about mid-infrared spectroscopy and waveguides has been added.
Point 5. The logic of some parts of the manuscript is a bit confusing, for example, the content of section 2.3 does not match the title of chapter 2(Fundamental Principles: Spectrophotometric Evaluation on Chip).
Response 5: In the revised manuscript contents are well structured following reviewers’ suggestions. These sections have been condensed to give a brief introduction to common readers and title has been changed.
Point 6. Some applications of disease diagnostics for the MIR thin-film waveguide need to be supplemented in the description of waveguide materials. (chapter 4 of the manuscript)
Response 6: The applications of disease diagnostics for the MIR thin-film waveguide has been added (page 34 lines 13-23 and pages 35, 36, 37, 38) and supported by Fig. 9 and Fig. 10.
Point 7. In section 5, the incorporation of graphene into waveguides is not discussed thoroughly. Please add the relevant content.
Response 7: The incorporation of graphene into waveguides has been added, in the revised manuscript (page 45 line 14 to 23 and page 46 line 1 to 14).
Please find attachment

Round 2
Reviewer 1 Report
the current version has been well-revised and ready for acceptance.
Reviewer 2 Report
The manuscript has been improved enough to be accepted for publishing in Sensors.
Reviewer 3 Report
I think the authors have addressed the issues well.